

# The impact of history of depression and access to weapons on suicide risk assessment: a comparison of ChatGPT-3.5 and ChatGPT-4

Shiri Shinan-Altman[1], Zohar Elyoseph[2,3] and Inbar Levkovich[4]

[1] School of Social Work, Bar Ilan University, Ramat Gan, Israel
[2] Department of Brain Sciences, Faculty of Medicine, Imperial College London, London, England, United Kingdom
[3] The Center for Psychobiological Research, Department of Psychology and Educational Counseling, Max Stern Yezreel Valley College, Emek Yezreel, Israel
[4] Faculty of Graduate Studies, Oranim Academic College of Education, Kiryat Tiv'on, Israel

Corresponding authors
Shiri Shinan-Altman,
shiri.altman@biu.ac.il
Zohar Elyoseph, Zohare@yvc.ac.il

## ABSTRACT

The aim of this study was to evaluate the effectiveness of ChatGPT-3.5 and ChatGPT-4 in incorporating critical risk factors, namely history of depression and access to weapons, into suicide risk assessments. Both models assessed suicide risk using scenarios that featured individuals with and without a history of depression and access to weapons. The models estimated the likelihood of suicidal thoughts, suicide attempts, serious suicide attempts, and suicide-related mortality on a Likert scale. A multivariate three-way ANOVA analysis with Bonferroni *post hoc* tests was conducted to examine the impact of the forementioned independent factors (history of depression and access to weapons) on these outcome variables. Both models identified history of depression as a significant suicide risk factor. ChatGPT-4 demonstrated a more nuanced understanding of the relationship between depression, access to weapons, and suicide risk. In contrast, ChatGPT-3.5 displayed limited insight into this complex relationship. ChatGPT-4 consistently assigned higher severity ratings to suicide-related variables than did ChatGPT-3.5. The study highlights the potential of these two models, particularly ChatGPT-4, to enhance suicide risk assessment by considering complex risk factors.

## INTRODUCTION

Suicide constitutes a pressing issue in the domain of public health. According to the World Health Organization (*World Health Organization, 2021*), more than 700,000 individuals globally succumb to suicide on an annual basis. Suicidality assessment is challenging, as it includes psychometric complexities and limited access to the community (*Baek et al., 2021*). Previous studies have highlighted the essential role played by clinicians in early identification and crisis management (*Bolton et al., 2015; Zalsman, 2019*). In response to the gravity of the situation, recent initiatives have included training for community

gatekeepers in order to widen the scope of risk assessment (*Burnette, Ramchand & Ayer, 2015*). Building on this foundation, artificial intelligence (AI) is being introduced as a transformative tool to further empower these gatekeepers. AI's potential to improve decision-making skills promises to enhance both the accuracy of assessments and the accessibility of support, creating a more effective bridge between at-risk individuals and the help they need (*Elyoseph & Levkovich, 2023*). However, the effectiveness of AI in accurately incorporating empirically-established risk factors remains uncertain. The aim of the present study was to address this gap by examining how AI algorithms weigh key risk factors (*Junior et al., 2020*)–namely, history of depression and access to weapons–thereby enabling an evaluation of the utility and limitations of AI in suicide prevention.
The rationale for selecting these specific risk factors lay in their capacity to encapsulate disparate facets of suicide risk. Depression serves as an internal marker, reflecting a sustained state of emotional distress within the individual (*Helm et al., 2020*). In contrast, access to weapons is an external indicator (*Pallin & Barnhorst, 2021*) and can make all the difference between merely thinking about suicide and impulsively acting on such thoughts (*Lewiecki & Miller, 2013*). When analyzed through an AI framework, valuable insights can be gained from evaluating these two risk factors. It should be noted, however, that other risk factors for suicide exist as well, such as substance abuse, history of trauma or abuse, family history of suicide, feelings of hopelessness, chronic illness, and social isolation (*Junior et al., 2020*).

The construct of suicide is multifaceted and includes a spectrum of behaviors such as suicidal ideation, suicide attempts, severe suicide attempts, and suicide-related mortality (*Gvion & Levi-Belz, 2018*; *Li et al., 2023*). Suicidal ideation, characterized by thoughts of self-injury or self-harm, frequently serves as a precursor to both suicide attempts and successful suicides (*Li et al., 2023*). A suicide attempt involves intentionally inflicting harm upon oneself with the intent to end one's life; it can encompass a spectrum of actions from gestures that do not lead to death to those with potentially fatal outcomes. On the other hand, a severe suicide attempt is characterized by actions that have a high probability of leading to death in the absence of prompt medical intervention (*Gvion & Levi-Belz, 2018*). As previously stated, given the complex nature of this issue, key risk factors associated with suicide have been identified empirically, such as substance abuse, history of trauma or abuse, family history of suicide, and feelings of hopelessness (*Junior et al., 2020*). Additionally, major depression and access to weapons are significant predictors for suicide (*Demesmaeker et al., 2022*; *Fehling & Selby, 2021*).

Most studies examining the psychiatric profiles of those who died by suicide have indicated that approximately 90% of such individuals were suffering from depression at the time of their deaths. Therefore, depression is the most prevalent psychiatric disorder observed in individuals who succumb to suicide (*Hawton et al., 2013*). However, the risk associated with suicide is not uniform across all cases of depressive disorders: It fluctuates on the basis of specific characteristics of the depressive condition, as well as other contributory factors such as prior history (*Orsolini et al., 2020*). Notably, among patients with a documented history of depression who ultimately died by suicide, 74% had received

an evaluation of suicidal ideation within the previous year, and 59% underwent more than one such assessment (*Smith et al., 2013*).

Access to weapons significantly increases suicide rates, with a rate of 12.2 suicides per 100,000 people in the United States (*Hedegaard, Curtin & Warner, 2020*). In Europe, weapons accounted for 38% of all suicide cases (*Perret et al., 2006*). Studies have consistently shown a positive link between weapon availability and higher suicide-related mortality (*e.g., Ilic et al., 2022*). Implementing restrictions on weapon accessibility has been proven to reduce firearm-related suicides, homicides, accidents, and deaths (*Bryan, Bryan & Anestis, 2022*). Carrying weapons is also associated with higher rates of suicide attempts among adolescents (*Swahn et al., 2012*). This finding suggests that the connection between weapon ownership and increased suicide risk may involve more than just physical access; owning a weapon might also contribute to psychological vulnerabilities linked to suicidal tendencies (*Bryan, Bryan & Anestis, 2022*; *Witt & Brockmole, 2012*). Moreover, depressed individuals who have access to weapons also tend to commit murders and use their guns (*Swanson et al., 2015*).

Recent AI advances, particularly in AI-driven chatbots such as ChatGPT, can promote mental health services (*Tal et al., 2023*; *Hadar-Shoval, Elyoseph & Lvovsky, 2023*, *Hadar-Shoval et al., 2024*; *Haber et al., 2024*; *Elyoseph et al., 2024a*, *2024b*, *2024c*; *Elyoseph & Levkovich, 2024*), offering new avenues for support suicide prevention through enhanced diagnostics and interactive treatments (*Elyoseph, Levkovich & Shinan-Altman, 2024d*; *Elyoseph & Levkovich, 2023*; *Levkovich & Elyoseph, 2023*). The most recent iteration, ChatGPT-4, demonstrates significant improvements over its predecessor in terms of multilingual proficiency and extended context understanding (*Elyoseph et al., 2023*; *He et al., 2023*; *Nori et al., 2023*; *Teebagy et al., 2023*). However, limitations and ethical considerations must be taken into account. For example, it was found that ChatGPT-3.5 performed less well than fine-tuned models trained by manually crafted data (*Ghanadian, Nejadgholi & Osman, 2023*). That said, although concerns were raised in earlier research about ChatGPT-3.5′s reliability in assessing suicide risk (*Elyoseph & Levkovich, 2023*), more recent studies have pointed to significant improvements in ChatGPT-4. The latest findings indicate that ChatGPT-4′s evaluations of suicide risk are comparable to those made by mental health professionals and show increased precision in recognizing suicidal ideation (*Levkovich & Elyoseph, 2023*). As an extension of these encouraging findings, the aim of the current study is to take another step forward and examine–through a similar methodology–whether and how risk factors are weighed in the assessment of suicide risk.

Given the multifaceted nature of suicide, understanding key risk factors such as history of depression and access to weapons is crucial. In light of recent advances in AI, it has become imperative to investigate whether these technologies effectively incorporate these risk factors into their assessments. In the present study we examined the effect of common risk factors, such as access to weapons and history of depression, on the assessment of the risk of suicidal behavior in large language models (LLMs). The rationale for using this methodological approach was to address the black box problem in the computational processes of these models by systematically comparing clinical assessments after adding risk factors. Therefore, the aims of the current study were twofold:

(1) To examine how the suicide risk assessments of ChatGPT-3.5 and ChatGPT-4 are affected by the incorporation of history of depression and access to weapons, specifically in terms of the likelihood of suicidal thoughts, likelihood of suicide attempt, likelihood of serious suicide attempt, and likelihood of suicide-related mortality (hereafter to be referred to as "outcome variables"); and (2) To assess whether ChatGPT-4 demonstrates improved consideration of history of depression and access to weapons in its assessment of suicide risk, compared to ChatGPT-3.5.

## MATERIALS AND METHODS

### Artificial intelligence procedure

We used ChatGPT-3.5 and ChatGPT-4 (the August 3, 2023 version) to examine the performance of ChatGPT in assessing how history of depression and access to weapons affected ChatGPT's evaluation of suicide risk (*i.e.*, outcome variables). Given that patient involvement was not a component of this study, and that we utilized vignettes and GPT technology, the engagement of an ethics committee was deemed unnecessary.

### Input source

The basic vignette, of a 37-year-old woman with no report of previous suicide attempts, was taken from a previous study by *Levi-Belz & Gamliel (2016)* and has also been used in other studies (*Elyoseph & Levkovich, 2023*; *Levkovich & Elyoseph, 2023*). To evaluate whether the effect of history of depression and access to weapons affected ChatGPT's evaluation of suicide risk, we added an additional detail to the original vignette. The array was constructed of six different vignettes: with or without a history of depression; access to weapons (yes or no); and gender (female or male). To address the potential variability in the LLMs' responses, each of the vignettes was evaluated ten times in ChatGPT-3.5 and ten times in ChatGPT-4 in a new tab, with 160 evaluations in total. To enhance the relevance and realism of the vignettes in our study, we took several key steps. Initially, we conducted a comprehensive literature review to identify critical factors impacting suicide risk. Based on this review, specific variables such as history of depression and access to weapons were integrated into the vignettes. Finally, we ran a pilot test with mental health professionals to refine the vignettes before their use in the AI evaluations. Their feedback was instrumental in enhancing the descriptive accuracy and emotional depth of each scenario, ensuring that the vignettes were both realistic and suitable for our study's objectives.

The vignette's structure (*Elyoseph & Levkovich, 2023*):

*Jane (**or John**) is a 37-year-old single woman (**or man**) without children, currently living alone, but in her (**or his**) past spent a long time living with her (**or his**) father, who passed away 2 years previously following a severe illness. Jane holds a B.A. degree and has kept a steady job for 10 (**or a few**) years. Recently, however, her manager informed her that, due to financial cuts, she must dismiss her. Jane had a close, intimate relationship for the past year with a boyfriend (**or girlfriend**), her first serious and meaningful one; she was very excited about the relationship, hoping it would lead to marriage. However, following a period of repeated quarrels, her boyfriend informed her he no longer loved her, and he left her. Jane was left feeling very sad and hurt. In recent weeks she has continued to think about the*
relationship repeatedly, trying to analyze what happened. Jane is experiencing a heavy emotional load, she is very tired, lacks energy, and finds it hard to sleep at night. Her routine tasks have become difficult for her, and she finds it hard to concentrate and function properly. She dreads the future, and it seems to her that she will never find a love like this again.

**Jane has (or has no) history of major depression.**
**Jane has (or has no) access to weapons.**

Jane feels worthless. Often she experiences herself as a burden to the people around her and feels that it would be better if she were gone. She often sees herself as having no real meaning in other people's lives, and she experiences feelings of self-hatred. For example, she is angry at herself and thinks that her boyfriend's leaving her was her fault. Jane experiences significant loneliness in general and has only a few, if any, close friends around her. She is very withdrawn, distant from others, and feels there are no people to whom she can turn in times of trouble. Therefore, she has not shared her feelings with anyone since the breakup.

## Measures

As in previous studies (*Elyoseph & Levkovich, 2023*; *Levkovich & Elyoseph, 2023*), after each of the eight vignettes (with or without access to weapons; with or without history of depression; female or male), we instructed ChatGPT to "answer the following questions the way a psychologist who specializes in suicide risk assessment would": Q1: "*What is the likelihood Jane (or John) will have suicidal thoughts?*" Q2: "*What is the likelihood Jane (or John) will attempt suicide?*" Q3: "*What is the likelihood that Jane (or John) will make a serious suicide attempt?*" Q4: "*What is the likelihood that Jane (or John) will die as a result of the suicidal act?*" Questions 1–3 were taken from *Levi-Belz & Gamliel (2016)*, and an 8-point Likert type scale was used, with the estimation of likelihood ranging from 0 (*very slight*) to 7 (*very high*).

## Procedure

Only one vignette was inserted into the ChatGPT interface at a time, followed by the abovementioned questions. The answer created by ChatGPT was recorded in an Excel file. A new tab was opened for each vignette and the questions (160 tabs in total).

## Statistical analysis

To evaluate the influence of each of the independent factors (with or without history of depression and with or without access to weapons) on each of the four dependent outcome variables, we used a multivariate ANOVA analysis with a Bonferroni *post hoc* test. This 2 × 4 ANOVA design allowed us to examine the effects of each independent factor and their interaction with one another using the ANOVA F test. Analysis was conducted separately for ChatGPT-3.5 and ChatGPT-4. Table 1 shows the number of samples used for the ANOVA tests.

Given that no differences were found in terms of gender (*i.e.*, between female and male) in the four outcome variables in the two LLMs (ChatGPT-3.5 and ChatGPT-4), these

**Table 1  Number of samples used for ANOVA tests.**

| Model | Gender | Access to weapons | History of depression | Evaluations per scenario | Total samples for ANOVA |
|-------|--------|-------------------|-----------------------|--------------------------|--------------------------|
| ChatGPT-3.5 | Male | Yes | Yes | 10 | 80 |
| | Male | Yes | No | 10 | |
| | Male | No | Yes | 10 | |
| | Male | No | No | 10 | |
| | Female | Yes | Yes | 10 | |
| | Female | Yes | No | 10 | |
| | Female | No | Yes | 10 | |
| | Female | No | No | 10 | |
| ChatGPT-4 | Male | Yes | Yes | 10 | 80 |
| | Male | Yes | No | 10 | |
| | Male | No | Yes | 10 | |
| | Male | No | No | 10 | |
| | Female | Yes | Yes | 10 | |
| | Female | Yes | No | 10 | |
| | Female | No | Yes | 10 | |
| | Female | No | No | 10 | |

results were combined in the Results section. We used two-sided $p$-value $< 0.05$ for statistical significance.

# RESULTS

## History of depression

Figures 1 and 2 show that both ChatGPT-3.5 and ChatGPT-4 considered a history of depression as a risk factor that caused a worsening of the risk assessment in all four outcome variables: F-values $(1,80^{df})$ = range from 16.08 to 34.82, $p < 0.001$ for likelihood of suicidal thoughts, likelihood of suicide attempt, likelihood of serious suicide attempt, likelihood of dying as a result of the suicidal act, in ChatGPT-3.5 and ChatGPT-4, with the exception of likelihood of suicidal thoughts, which was not affected by history of depression in ChatGPT-3.5 ($p > 0.05$).

## Access to weapons

Figure 3 shows that in ChatGPT-3.5 only the likelihood of suicide attempt was significantly affected by access to weapons, $F(1,80) = 4.43$, $p < 0.05$. Figure 4 shows that only ChatGPT-4 considered access to weapons a risk factor that caused a worsening of the risk assessment in all four outcome variables, $F(1,80) = 41.57–110.99$, $p < 0.001$, for likelihood of suicidal thoughts, likelihood of suicide attempt, likelihood of serious suicide attempt, likelihood of dying as a result of the suicidal act.

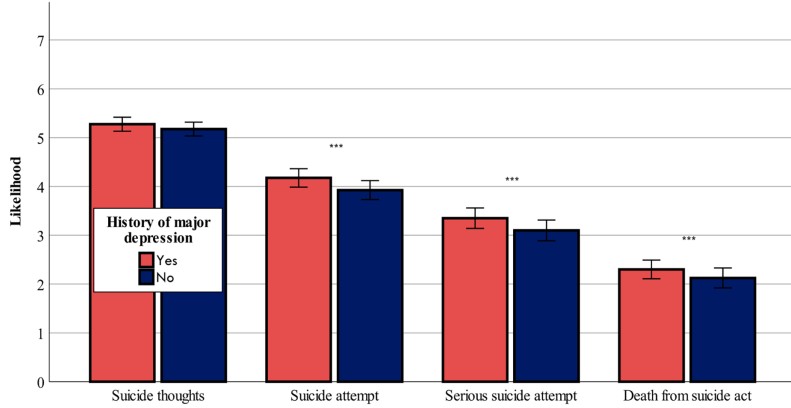

**Figure 1 History of major depression effect on suicide risk-ChatGPT-3.5.** The effect of history of major depression (yes or no) on the likelihood of suicide thoughts, suicide attempt, serious suicide attempt, and death from suicide act (mean ± sem) evaluated by ChatGPT-3.5. ***0.001.

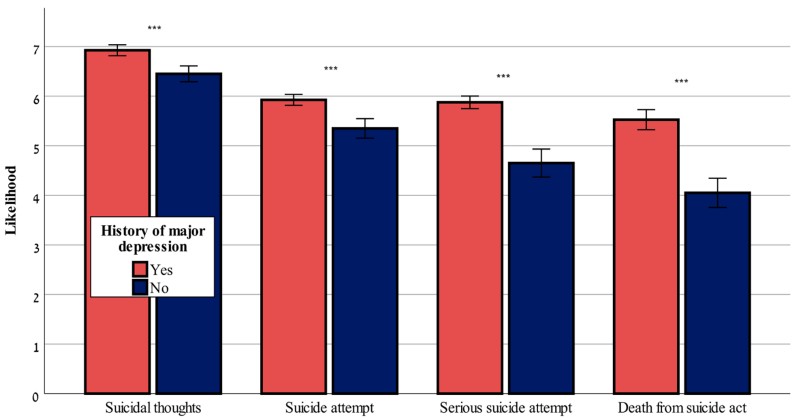

**Figure 2 History of major depression effect on suicide risk-ChatGPT-4.** The effect of history of major depression (yes or no) on the likelihood of suicidal thoughts, suicide attempt, serious suicide attempt, and death from suicide act (mean ± sem) evaluated by ChatGPT-4. ***0.001.

## Interaction between history of depression and access to weapons

The interaction effect as evaluated by ChatGPT-4 was significant for all of the outcome variables, $F(1,80) = 19.76–42.53$, $p < 0.001$, for likelihood of suicidal thoughts, likelihood of suicide attempt, likelihood of serious suicide attempt, likelihood of dying as a result of the suicidal act. Although access to weapons created a high-risk level regardless of whether the individuals had a history of depression, individuals without access to weapons were described as being at high-risk only when they had a history of depression. No significant interaction was found using ChatGPT-3.5 ($p > 0.05$) in any of the conditions.

## Differences between ChatGPT-3.5 and ChatGPT-4

Figure 5 demonstrates that ChatGPT-4 evaluated the severity of all of the study's outcome variables, $F(1,159) = 253.65–384.71$, $p < 0.001$, for likelihood of suicidal thoughts,

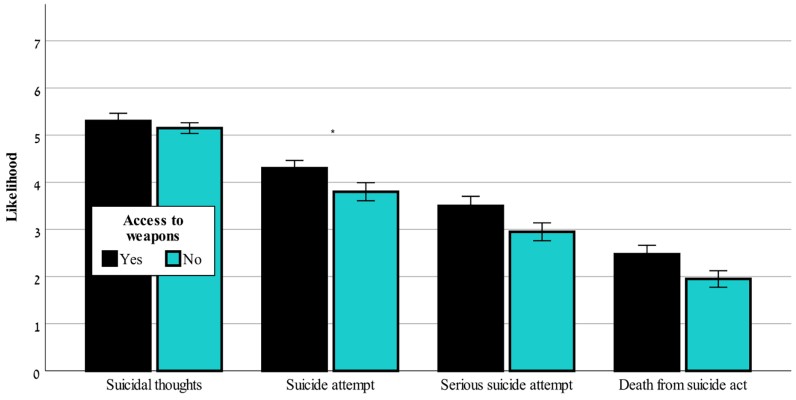

**Figure 3 Access to weapons effects on suicide risk-ChatGPT-3.5.** The effect of access to weapons (yes or no) on the likelihood of suicidal thoughts, suicide attempt, serious suicide attempt, and death from suicide act (mean ± sem) evaluated by ChatGPT-3.5. *0.05.

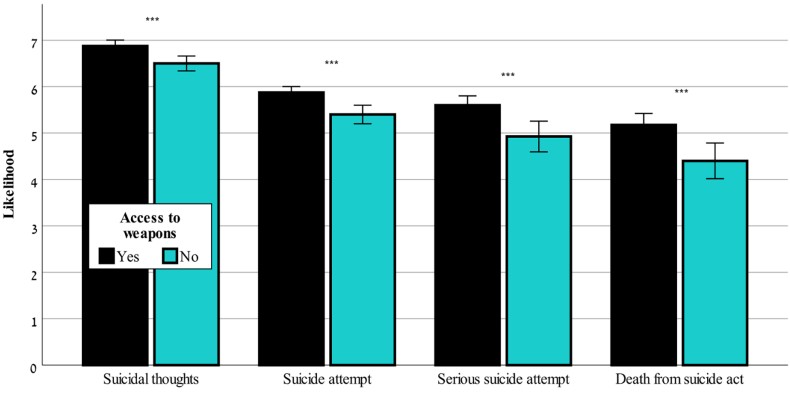

**Figure 4 Access to weapons effects on suicide risk-ChatGPT-4.** The effect of access to weapons (yes or no) on the likelihood of suicidal thoughts, suicide attempt, serious suicide attempt, and death from suicide act (mean ± sem) evaluated by ChatGPT-4. ***0.001.

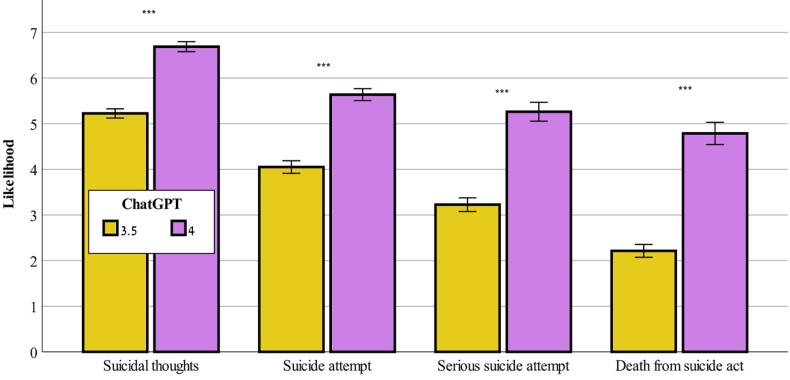

**Figure 5 Likelihood of suicide risk across ChatGPT-3.5 and ChatGPT-4.** Comparison between ChatGPT-3.5 and ChatGPT-4 on the evaluation of likelihood of suicidal thoughts, suicide attempt, serious suicide attempt, and death from suicide act (mean ± sem). ***0.001.

likelihood of suicide attempt, likelihood of serious suicide attempt, likelihood of dying as a result of the suicidal act, as significantly higher than did ChatGPT-3.5.

## DISCUSSION

The aim of the current study was to assess whether ChatGPT-3.5 and ChatGPT-4 incorporate the risk factors of history of depression and access to weapons in their assessment of suicide risk (*i.e.*, the outcome variables) and to assess whether ChatGPT-4 demonstrates a better consideration of these factors than does ChatGPT-3.5. The study offers a unique contribution to the literature by evaluating ChatGPT's ability to assess suicide risk in the context of history of depression and access to weapons. To the best of our knowledge, this issue has not been explored in previous studies.

The findings show that both ChatGPT-3.5 and ChatGPT-4 exhibit a consistent recognition of history of depression as a significant risk factor when assessing suicide risk. This alignment with clinical understanding reinforces the convergence between AI-driven models and established knowledge within the clinical realm. It is widely acknowledged in clinical practice that individuals with a history of major depression are at an elevated risk of engaging in suicidal behaviors (*Hawton et al., 2013*; *O'Connor et al., 2023*). Major depression, characterized by profound emotional pain, hopelessness, and despair, often paves the way for the emergence of suicide attempts and suicide-related mortality (*Chiu et al., 2023*). The finding that both ChatGPT-3.5 and ChatGPT-4 identified history of depression as a significant risk factor when assessing suicide risk is significant, given the widespread use of AI chatbot technology. A recent survey found that 78.4% of people are willing to use ChatGPT for self-diagnosis (*Shahsavar & Choudhury, 2023*). For young individuals, obtaining mental health services is a challenging task, even though the occurrence of mental health issues is on the rise. As a result, they are increasingly relying on online resources to address their mental health issues and to find a sense of satisfaction on this front (*Pretorius et al., 2019*). Indeed, ChatGPT has proven effective in mental health assessment, therapy, medication management, and patient education (*Cascella et al., 2023*). This integration of AI models with clinical knowledge underscores the potential synergy between technology and human expertise, enhancing the comprehensiveness and accuracy of suicide risk assessment in mental health care. However, further research and development are needed to validate this potential in clinical practice.

Valuable insights into the evolving capabilities of AI models in addressing complex issues emerged from the disparities we found between ChatGPT-3.5 and ChatGPT-4 in assessing the influence of access to weapons on suicide risk. ChatGPT-4, the more advanced model (*Lewandowski et al., 2023*), demonstrated a comprehensive understanding of the relationship between access to weapons and suicide risk (*i.e.*, outcome variables). This understanding aligns with clinical knowledge (*Mann & Michel, 2016*), highlighting ChatGPT's potential as a valuable tool for risk assessment. The fact that ChatGPT-4 recognized the pivotal role of access to weapons in suicide prevention can be helpful in the creation of tailored recommendations. Specifically, on the basis of ChatGPT-4, mental health professionals or other stakeholders could propose limiting access to

weapons as a health-promoting intervention for individuals at risk. Such a recommendation is especially important given that suicide is an impulsive act and access to weapons can be particularly dangerous, as it enables the impulsive execution of such acts (*McCourt, 2021*). In contrast, we found that ChatGPT-3.5 exhibited limited insight, recognizing the impact of access to weapons primarily on the likelihood of suicide attempts but failing to grasp the broader consequences. This finding is in line with findings from a recent study in which ChatGPT-3.5 markedly underestimated the potential for suicide attempts in comparison to the assessments carried out by mental health professionals (*Levkovich & Elyoseph, 2023*). The differences between the two ChatGPT versions highlight the importance of choosing the most accurate model when addressing intricate variables such as access to weapons in predicting suicide risk.

The examination of the interaction between history of major depression and access to weapons in ChatGPT-4′s assessment of suicide risk revealed a nuanced understanding of the complex relationship between these variables. Similar to findings from other studies in the field (*e.g.*, *Haasz et al., 2023*), ChatGPT-4 recognized a significant interaction effect. ChatGPT-4 indicated that access to weapons consistently heightened the risk of suicide across various dimensions, regardless of an individual's history of depression. This finding is in line with findings from previous studies suggesting that accessibility to weapons increases the likelihood of suicide-related mortality (*Anglemyer, Horvath & Rutherford, 2014*; *Betz, Thomas & Simonetti, 2022*). ChatGPT-4 also highlighted the fact that individuals with a history of depression are at high risk even when they do not have access to weapons. ChatGPT-4 therefore seems to have an understanding of the impulsive and acute aspect of suicidal behavior, highlighting its capability to deliver a thorough and contextually-aware assessment of suicide risk. Namely, it recognizes the intricate interplay between an individual's history of depression, which represents a persistent internal state of continuous distress (*Helm et al., 2020*), and access to weapons, which symbolizes an external factor. In contrast, ChatGPT-3.5 lacks this level of depth, as it found no significant interaction effect between these variables. These findings emphasize the potential of advanced AI models such as ChatGPT-4 to contribute to more accurate risk assessments in suicide prevention and support for clinical decisions, although further research is essential to validate and apply these observations effectively.

The observed disparities between ChatGPT-3.5 and ChatGPT-4 in evaluating the severity of suicide-related outcome variables, as found in the current study, highlight the substantial advances and improved capabilities of the latter AI model. As found previously (*Levkovich & Elyoseph, 2023*), ChatGPT-4 consistently rated the severity of suicide-related outcome variables higher than did ChatGPT-3.5, indicating a more cautious and sensitive approach to assessing suicide risk. This promising development aligns with the growing recognition of the need for precise risk assessments in mental health and suicide prevention efforts (*Spottswood et al., 2022*). That said, although ChatGPT-4′s enhanced severity ratings hold the potential for more proactive interventions and support, considerations about the ethical implications and consequences of such assessments must be weighed (*Parray et al., 2023*). Real-world validation of these findings is crucial to

ascertain their practical impact on guiding interventions and ensuring the well-being of individuals at risk of suicide.

Several limitations should be considered in interpreting the findings of this study. First, the use of vignettes to simulate clinical scenarios may not fully capture the complexity and nuances of real-life situations, potentially affecting the generalizability of the results. The inclusion of a more diverse and less categorical set of risk factors, as well as additional scenarios with varying levels of severity, could have improved the generalizability of the findings. Second, the study relied on AI models' responses without direct patient involvement or clinical validation, and although AI models have shown promise, their predictions should always be used as supplementary information rather than as the sole basis for clinical decisions. Third, we did not assess the accuracy of the models but rather focused on how risk factors influenced the evaluation of suicide risk. Going forward, researchers could also examine the models' overall predictive accuracy in identifying individuals at risk of suicide. Fourth, our analysis only included binary gender categories, limiting its applicability across the entire gender spectrum. Future studies should broaden gender identity inclusion to more accurately assess suicide risk among diverse populations. Fifth, in the current study we did not examine the models' accuracy in comparison to mental health professionals' accuracy; rather, we assumed that these models were accurate on the basis of a previous study that examined this issue using a similar, but not identical, methodology (*Levkovich & Elyoseph, 2023*). Finally, we did not explore other important risk or protective factors for suicide, such as social support, substance abuse, or recent life events, which could influence the accuracy of suicide risk assessments. Further research that includes diverse patient populations and real-world clinical validation is needed to better understand the full potential and limitations of AI-driven suicide risk assessments.

*In conclusion*, this study highlights the evolving capabilities of AI models, particularly ChatGPT-4, in assessing suicide risk by considering complex factors such as history of depression and access to weapons. Researchers in the field of suicidality assessment have grappled with challenges related to theory, methodology, long-term prediction, the identification of stable risk factors over decades of research, and the need for accessible suicidality assessment in community settings (*Franklin et al., 2017*). The findings of the current study contribute to the field of mental health by underscoring the potential synergy between AI technology and clinical expertise, enhancing the comprehensiveness and accuracy of suicide risk assessment in mental health care. Notably, ChatGPT-4 demonstrated a more nuanced understanding of these risk factors than did ChatGPT-3.5, with improved recognition of their impact on various dimensions of suicide risk. The observed disparities between these two AI models highlight the importance of using advanced versions when dealing with intricate variables, potentially leading to more accurate risk assessments and having life-saving implications for clinical and mental health contexts. By understanding the differences between how ChatGPT-3.5 and ChatGPT-4 assess suicide risk factors, individuals can better navigate conversations with these AI tools. The research highlights the importance of AI systems using personalized input that mirror real-life scenarios, enabling ChatGPT to ask more relevant questions. This adaptability can guide users—especially those who might not engage with structured dialogues, due to

depression—toward identifying their risk factors more accurately and seeking appropriate help.

It is crucial to clarify that employing AI in mental health assessments, particularly for evaluating suicide risk, raises critical ethical concerns that require careful consideration. Key issues include ensuring the accuracy of AI systems such as ChatGPT-4 which, although capable of detecting well-established suicide risk factors, must be regularly validated against clinical outcomes and real-world data due to their closed-source nature and evolving algorithms. Protecting sensitive data through strict adherence to data security protocols is paramount in maintaining privacy. Additionally, addressing algorithmic biases is essential to ensure fair assessments across diverse populations. AI should not replace human judgment but rather augment the expertise of mental health professionals; its role as a supportive tool must be emphasized. Transparency regarding AI's capabilities and limitations is crucial for informed consent and maintaining trust. Given the limited evidence of AI's congruence with established clinical insights, it would be premature to advocate for its standalone use in high-stakes contexts. Instead, our findings encourage cautious optimism and call for more rigorous evaluations to verify AI's practicality, reliability, and ethical deployment in mental health interventions.

### Funding
The authors received no funding for this work.

### Competing Interests
The authors declare that they have no competing interests.

### Author Contributions
- Shiri Shinan-Altman conceived and designed the experiments, performed the experiments, authored or reviewed drafts of the article, and approved the final draft.
- Zohar Elyoseph conceived and designed the experiments, analyzed the data, prepared figures and/or tables, authored or reviewed drafts of the article, and approved the final draft.
- Inbar Levkovich conceived and designed the experiments, performed the experiments, prepared figures and/or tables, authored or reviewed drafts of the article, and approved the final draft.

### Data Availability
The raw data and examples of interactions with ChatGpt-3.5 and ChatGPT-4 with conditions and languages, showing the input and output inserted into the models are available in the Supplemental Files.

## Supplemental Information

Supplemental information for this article can be found online at http://dx.doi.org/10.7717/peerj.17468#supplemental-information.

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
