# Peer review of "The impact of history of depression and access to weapons on suicide risk assessment: a comparison of ChatGPT-3.5 and ChatGPT-4"

_PeerJ, doi:10.7717/peerj.17468_

## Round 0.1 · original submission · Minor Revisions

Thanks for your submission. I have reviewed all the four reports from the referees' (two of them have recommended "minor changes", and Reviewers 3 and 4 have recommended "major changes"). If you note carefully their concerns, Reviewer 3 has questions and concerns about additional references and a run of the model check (assessment of accuracy), and Reviewer 4's comments are around the style and clarity of writing mostly. So, overall, my assessment is that they are asking really for relatively minor changes that. after reading your paper, I am confident, you will be able to address.

If you can please respond to these concerns say within the next two to three weeks and resend, I think we will be in a position to take a positive decision.

**Language Note:** The review process has identified that the English language must be improved. PeerJ can provide language editing services - please contact us at [email protected] for pricing (be sure to provide your manuscript number and title). Alternatively, you should make your own arrangements to improve the language quality and provide details in your response letter. – PeerJ Staff

Reviewer 1 ·

Basic reporting

no comment

Experimental design

- In the statistical analysis section, please indicate what criteria were used to determine a statistical significance. For example, a two-sided p-value < 0.05 indicates statistical significance.
- In line 156, you described that likelihood of suicidal thoughts, likelihood of suicide attempt, likelihood of serious suicide attempt, and likelihood to die from the suicide act as independent variables. I think these four variables are the outcome variables in your study So they should be dependent variables?
- In line 153, it’s a bit unclear how you conducted the ANOVA analysis. I understand that you conduct the ANOVA test for each of the four outcome variables (likelihood of suicidal thoughts, likelihood of suicide attempt, likelihood of serious suicide attempt, and likelihood to die from the suicide act). Could you clarify what the independent factors are here? Are they with vs. without a history of depression and with vs. without access to weapons? Did you conduct the ANOVA test for these two independent variables separately? If yes, I don’t think you need ANOVA since ANOVA is used for comparison between over two groups. Two sample t-test, or Wilcoxon rank-sum test would be more appropriate here.
- In line 178, you described the results of assessing whether there was the interaction between history of depression and access to weapons. It’s unclear what statistical analyses were used for this interaction test. Could you describe it in your statistical analysis section?
- In line 164, in F(1,80) = 16.08-34.82, please indicate what each value (1, 80, 16.08, and 34.82) means.

Validity of the findings

no comment

Additional comments

no comment

·

Basic reporting

1. Refer to line 41-43.
2. Refer line 82-84 (ANONYMISED1).
3. Refer (ANONYMISED2) Line 114
4. Explain ‘Some medical knowledge’ in line 77 or change this phrase.
5. The author needs to refer to ‘The construct of suicide’ in line 44. Also, explain the difference between suicide attempts and severe suicide attempts.

Experimental design

1. The authors addressed suicide attempts, depression, and access to weapons; but, they also had to discuss the fact that depressed individuals who have access to weapons also tend to commit murders and gunfire. Justify.
2. The most recent and advanced iteration of the ChatGPT AI bot is ChatGPT 4.5. Although this version was released in April 2023, why didn't the author use it?
3. Results are not enough to justify the objective of the study. The authors need to add more factors to the methodology to validate the study's aims and objectives.

Validity of the findings

The provided data are statistically sound and conclusions are well-stated.

Reviewer 3 ·

Basic reporting

This paper investigates whether GPT systems are sensitive to two risk factors, namely, history of depression and access to weapons, in determining the severity of suicide risk. The paper is well-written and organized. It is easy to follow and understand.

Overall, the background is comprehensive. However, the paper misses some relevant references. For example, this work misses the points raised by Ghanadian et al. (https://aclanthology.org/2023.wassa-1.16/), which show that GPT3.5 underperforms fine-tuned models trained by manually crafted data. They also reveal that the temperature parameter is essential when assessing GPT models.

Experimental design

The research is within the scope of the journal. The research question is well-defined. Results are presented. The authors have a good knowledge of the field.

Some details are missing from the experiments. I suggest a table clearly stating the number of samples used for ANOVA tests. In my understanding, the sample size is too small, but that might be resolved after clarification.

Also, the paper looks at the model's sensitivity to risk factors by assessing the predicted likelihood of severity when the input changes. This sensitivity is only meaningful if the model correctly predicts the severity. However, the paper fails to report the accuracy of each model. In this scenario, an extreme example would be a model that labels everything highly severe when a risk factor is present. Of course, that model shows high sensitivity to this risk factor, but that might come at the cost of high false alarms. That model is not an ideal one. From the results of this paper, we don't have this information, which is a major methodological flaw. Adding this information to the paper is necessary.

Another methodological flaw is ignoring the role of temperature parameters. GPT systems only generate reproducible results if the temperature is set to zero through the API. With other temperature values, the results contain uncertainty, and that is not recommended for a clinical setting.

Validity of the findings

I suggest the authors discuss the impact of the results in more specific terms. In the conclusion section, the authors seem to recommend that GPT models are practical and valuable. However, their results show these models are sensitive to two well-known risk factors. Given that these models are closed-source and constantly changing, even the most accurate GPT models need to be evaluated regularly, and limited evidence of alignment with clinical knowledge is not enough to recommend these systems for critical tasks such as suicide attempt detection.

Additional comments

Line 51: typo - complex nature
There is no need for anonymized references as authors' names are known.
I suggest the authors mention that they are considering only binary gender as one of the limitations of their work.

·

Basic reporting

Please use correct grammar in line 51. Refer to the word “complexity nature’, maybe replace it with ‘complex nature’.
The shift from the topic of suicide to AI’ use is abrupt and sudden, please build a more connected narrative here. Refer to line 32.
Please cite a reference for mentioning Depression and access to weapons as key factors. Refer to line 36.
Please mention other risk factors in the introduction for suicide as well so that the reader can also understand the topic clearly.
Please cite a reference or change the structure of the sentence in line 42 and 43.
Please mention other factors along with Depression and Access to weapons. Refer to line 52 and 53.
Please start line 54 with the sentence in line 55 “A majority of studies in…” and then state the sentence from line 54 ‘Depression is the most prevalent…” by adding a therefore in the start of this sentence. Example, “Therefore, Depression becomes the most prevalent…”.
In line 63 and 64, the reference is cited for the US only, is your study only based on evaluating suicide based on depression and access to weapons in the US only? Please use a more globally applicable standard here.
In line 81, before the reference (Elyoseph & Levkovich, 2023), please elaborate which ‘crucial factor’ are you referring to.
Please also try to make the introduction section a bit more concise, this will help the readers to relate the background with the objective of the study much quickly and easily.
Please work on the formatting of figures and their details below the figures. Please keep the formatting consistent throughout the paper.

Experimental design

Please elaborate more on how the chat GPT technology is used in the section Material and Methods in line 99.
In line 110 to 112, please mention all 8 different vignettes.
In line 112-113, what is the rationale behind evaluating the vignettes 10 times? Please elaborate on this more in your paper.
In the section Measures, in line 145, question number 4 is not logically correct or can be answered by anyone with any measure of certainty. Please reconsider this question in your research.

The overall design and methodology need to be revisited in order to make the research question more logically correct. Please try to investigate the research question more by using real-life data and not only from vignettes.

Validity of the findings

The research question was to evaluate a difference of suicide risk factors between ChatGPT 3.5 and 4 and also to find out if these two LLMs consider the factors as a basis of suicide. However, the use of the study should also highlight how an individual can make use of such findings? A depressed individual will not use vignettes, and his/her conversation with ChatGPT will be highly variable in terms of the inputs. The study should also use personalized inputs based on more real-life scenarios to explore if ChatGPT is able to ask the right questions/inputs from the users or not.

---

## Round 0.2 · Minor Revisions

Congratulations! We have provisionally accepted your submission subject to the following two recommendations from our reviewer:
1. Provide a more detailed description of vignette selection and construction to ensure replicability and discuss the ethical implications of using AI for mental health assessments.
2. Improve grammar and style for clarity and professionalism, ensure consistent terminology throughout the paper, and update and standardize reference formatting.
Well done!

Reviewer 1 ·

Basic reporting

The authors have sufficiently addressed my comments.

Experimental design

The authors have sufficiently addressed my comments.

Validity of the findings

The authors have sufficiently addressed my comments.

·

Basic reporting

The paper is well-organized and offers a clear narrative that effectively highlights the differences between the AI models in a clinical context. While, I have outlined some specific areas where enhancements could be made to further strengthen your submission.

1. Provide a more detailed description of vignette selection and construction to ensure replicability and discuss the ethical implications of using AI for mental health assessments.
2. Improve grammar and style for clarity and professionalism, ensure consistent terminology throughout the paper, and update and standardize reference formatting.

Experimental design

The methodology section is thorough and provides a solid foundation for the study.

Validity of the findings

The discussion aligns well with the results, offering thoughtful interpretation and implications for future research and practice.

Additional comments

no comments

---

## Round 0.3 · accepted · Accept

Congratulations! Well done